# Preparation of Butadiene-Isoprene Copolymer with High Vinyl Contents by Al(OPhCH_3_)(i-Bu)_2_/MoO_2_Cl_2_∙TNPP

**DOI:** 10.3390/polym11030527

**Published:** 2019-03-20

**Authors:** Peipei Li, Kai Liu, Zhe Fu, Yongliang Yu, Zhaobo Wang, Jing Hua

**Affiliations:** 1Key Laboratory of Rubber-Plastics, Ministry of Education/Shandong Provincial Key Laboratory of Rubber-Plastics, Qingdao University of Science and Technology, Qingdao 266042, China; lppnjzy1011@163.com (P.L.); fuzhe95@163.com (Z.F.); 2Beijing National Laboratory for Molecular Sciences, Organic Solid Laboratory, Institute of Chemistry, Chinese Academy of Sciences, Beijing 100190, China; liukai91@iccas.ac.cn; 3Department of Chemistry, University of Chinese Academy of Sciences, Beijing 100049, China; 4College of Material Science & Engineering, Qingdao University of Science & Technology, Qingdao 266042, China; yongliangyu2013@163.com (Y.Y.); wangzhb@qust.edu.cn (Z.W.)

**Keywords:** coordination polymerization, Mo-based catalytic system, high vinyl structures, butadiene-isoprene copolymers

## Abstract

In this study, a butadiene-isoprene coordination polymerization was initiated by a binary molybdenum (Mo)-based catalytic system consisting of modified MoO_2_Cl_2_ as the primary catalyst, triethyl aluminum substituted by m-cresol as the co-catalyst and tris(nonyl phenyl) phosphate (TNPP) as the ligand. The effects of the amount of catalyst and type of co-catalyst were investigated in detail. Experimental results indicated that when the butadiene-isoprene coordination polymerization was initiated by the binary Mo-based catalytic system, the monomer conversion could reach 90%. The resulting butadiene units were primarily based on 1,2-structures, and the reactivity ratios of butadiene and isoprene were 1.13 and 0.31, respectively. The reaction in the catalytic system was attributed to the non-ideal and non-constant ratio copolymerization. When the addition of isoprene monomers was relatively low, the isoprene units on the butadiene-isoprene copolymers were primarily based on the 1,2- and 3,4-structures. Moreover, the orientation of active centers to 1,2- and 3,4-structures gradually decreased with an increase in the addition of isoprene monomers, which resulted in the generation of high vinyl butadiene-isoprene copolymers.

## 1. Introduction

Nowadays and into the future, low-cost, environmentally friendly and high-efficient materials are an enormous challenge in the material field. Rubber materials have aroused the intensive interest and attention from academia and industry due to many outstanding performances such as superior wet skid resistance, excellent fatigue resistance and low rolling resistance [1,2,3,4]. Among them, polybutadiene (PBd) and polyisoprene (PI) are regarded as two types of significant materials in the rubber industry, especially for tire manufacturing [5,6,7,8]. PI possesses superb aging resistance, high wet skid resistance, low heat build-up, and can be applied as the rubber material for high-performance tire tread, accounting for why there exists more 1,2- and 3,4-structures, thereby resulting in the decrease in main-chain double bonds and in the increase of vinyl group content [9,10,11,12]. High vinyl polybutadiene rubber (HVBR) with vinyl side group content more than 80% is prepared by the coordination polymerization with a Mo-based (molybdenum) catalyst as the catalytic system. In view of high insertion rate of vinyl groups, the increasing intermolecular distance, the decreasing abrasion among molecular chains and the low density, which is beneficial in making the material light and heat generation low, HVBR has developed as the optimum material for high-performance green tires [13,14]. Meanwhile, in comparison to natural rubber (NR), butadiene-styrene rubber (SBR) and butadiene rubber (BR), HVBR is endowed with excellent aging-resistant performance, superb wet skid resistance, superior damping and physical mechanical performances due to few main chain double bonds and dense side vinyl groups, which is favorable to enhance the using horizontal and life of tire [15,16,17]. Block copolymer composites are considered as perspective functional materials in various application fields. Self-assembled block copolymers (BCP) stand out as a platform for developing the next-generation high-performance membranes [18,19]. The copolymerization can not only regulate the content of vinyl structure on the molecular chain, but also introduce the isoprene structural units on the molecular chain and obtain butadiene-isoprene copolymers composed of different structural units, thereby pushing the restriction on existing rubber grades, expanding the application range of homopolymers and copolymers, and providing a more novel and valuable performance from the polymers.

There appears a certain difference in the coordination polymerization mechanism of butadiene and isoprene, thereby making the butadiene-isoprene copolymerization with high orientation difficult to conduct. When performing diolefin coordination polymerization, there exists eight modes for the coordination between diolefin monomers and metal, including double-seat coordination such as trans-1,4 and cis-1,4 coordination and single-seat coordination such as 1,2-re, 1,2-si, and others. Therefore, the high 1,2- and 3,4-selective copolymerization of conjugated dienes needs to be explored in detail, from theoretical prediction to experimentation through control of the electronic effects and steric resistance of substituents close to the metal active center atoms. However, in the case of butadiene and isoprene monomers, the electronic effect and steric resistance are different, not only resulting in the activity and orientation of the active centers to butadiene and isoprene monomers differing from each other, but also causing the reactivity ratios to be much more different. Consequently, on the basis of controlling trans-1,4, cis-1,4 and 1,2- (3,4-) regioselectivity and stereoselectivity of conjugated dienes, the steerable adjustment of the copolymer composition and distribution is one of the perpetual topics for the polymer coordination polymerization [20,21,22,23].

Butadiene-isoprene copolymerization has, thus far, been extensively researched. The synthesis of butadiene-isoprene copolymers can adopt different Ziegler-Natta catalytic systems, mainly including transition metals (titanium, cobalt, nickel, etc.), lithium and rare-earth metal complexes. Due to the numerous advantages over conventional transition metal catalysis, organo-catalytic methodologies have become an attractive synthetic tool for asymmetric catalysis [24,25]. More efforts about the research on butadiene-isoprene copolymers were currently devoted to the rare-earth catalytic system. The rare-earth catalytic system demonstrates high superb orientated polymerization ability, and achieves high molecular weight polymers among aliphatic hydrocarbons. There appear to be more cis-1,4 structures (>95%) on the resulting polymer molecular chains [26,27,28]. Y. Hu et al. [5] and Tanaka et al. [29] fabricated butadiene-isoprene copolymers with a high molecular weight distribution, low molecular weight and high cis-1,4 structure content by employing an Nd-based catalytic system to initiate the copolymerization. He et al. [30] employed a TiCl_4_/MgCl_2_-Al(i-Bu)_3_ catalytic system to initiate the butadiene-isoprene copolymerization. Results demonstrated that as the polymerization time increased, the structure of copolymers was primarily based on cis-1,4 units in the initial stage, and then the mole ratio of cis-1,4/trans-1,4 gradually increased, and ultimately, the trans-1,4 unites dominated on the molecular chains. D.C.D Nath [31] investigated a cobalt-based catalytic system consisting of cobalt dichloride with methyl aluminoxane and triphenylphosphine (P(Ph_3_)_3_), by adjusting the polymerization activity of the catalytic system through control of P(Ph_3_)_3_ addition. They discovered that the butadiene-isoprene copolymers initiated by the cobalt-based catalytic system could achieve reactivity ratios of *r*_1_ = 2.5 and *r*_2_ = 0.8. However, only limited investigations have focused on the high 1,2- and 3,4-stereoselective copolymerization of butadiene and isoprene monomers. Fortunately, Mo-based metal catalytic systems can selectively initiate the butadiene-isoprene copolymerization with high activity, high 1,2- and 3,4-structures [4,32,33,34], but more efforts are required to focus on the ligand selection for the butadiene-isoprene copolymerization. Although the operation of the solution polymerization is complex, this method makes the viscosity of the polymer system low, as well as reduces the gel effect, avoids the partial overheating, and lowers the risk coefficient for reaction.

In this research, the effects of different types of ligands on the catalytic activity and microstructures of the resulting copolymers were explored in detail. Experimental results showed that the butadiene-isoprene copolymerization could achieve high 1,2- and 3,4-stereoselective butadiene-isoprene copolymers using a binary molybdenum (Mo)-based catalytic system, composed of the aging-modified MoO_2_Cl_2_ as the main catalyst, the m-cresol as a co-catalyst and tris(nonylphenyl)phosphite (TNPP) as a ligand. Meanwhile, the effects of dosage of various components in catalytic system on the butadiene-isoprene copolymerization were explored as well.

## 2. Experimental

### 2.1. Materials

Butadiene (Bd, polymerization grade, >99%) with a mass fraction over 90% was supplied by Sinopec Qilu Rubber Factory, Zibo, China. Isoprene (Ip, polymerization grade, >99%) was provided by Qingdao Icos New Materials Co., Ltd., Qingdao, China. Butadiene and isoprene were dried with 4 Å molecular sieves and, were distilled and purified two times for later use. MoO_2_Cl_2_ was purchased from Shanghai Aladdin Biological Technology Co., Ltd., Shanghai, China. Hydrogenated gasoline, supplied by Qilu Petrochemical (Zibo, China), was distilled and dehydrated by activated alumina at 70 °C before using. Toluene (analytical reagent, >99%) was purchased from Yantai Sanhe Chemical Reagent Co., Ltd. (Yantai, China) and cyclohexane (analytical reagent, >99%) was purchased from Tianjin Bodi Chemical Co., Ltd. (Tianjin, China). Both toluene and cyclohexane needed to be refluxed by sodium metal in nitrogen atmosphere for 1 h, distilled and then dried by activated alumina for later use.

### 2.2. Preparation of Mo-Based Catalytical System

The m-cresol-substituted alkyl aluminum was prepared according to the following procedure: A certain volume of AlEt_3_ was introduced into a reactor with an electromagnetic agitator under the protection of nitrogen atmosphere and then adding the m-cresol into the triethyl aluminum at a mole ratio of AlEt_3_:HOPhCH_3_ of 1:1 to prepare Al(OPhCH_3_)Et_2_ co-catalysts.

The TNPP-modified MoO_2_Cl_2_ was prepared according to the following procedure: MoO_2_Cl_2_ and ligand TNPP were first mixed at a certain proportion and then had n-hexane added to dilute to the desired concentration. Finally, the mixed solution was sealed and placed for a period of time to form the homogeneous solution, that is, the main catalyst.

### 2.3. Preparation of Butadiene-Isoprene Copolymer

The quantitative butadiene monomers and isoprene monomers were added to the polymerization pipe under the protection of a nitrogen atmosphere, and subsequently introducing the co-catalyst, ligand TNPP and the main catalyst into the polymerization pipe successively. The reaction was allowed to proceed at a certain temperature and time. After that, the polymerization reaction was terminated by adding a certain amount of terminators consisting of the ethanol and 1 wt % of 2,6-tert butyl-4 cresol solution. Finally, the resulting polymer was precipitated by adding enough ethanol at room temperature, and transferred to an oven at 60 °C for 4 h to remove any trace of residual solvent.

### 2.4. Measurements

The monomer conversion was calculated after thoroughly washing and precipitating the resulting polymer with an ethanol solution, which was then vacuum dried to a constant weight according to the following formula:
Con% = 100*W*t∙*W*_0_^−1^(1)
where *W*_0_ is the weight of butadiene and isoprene monomers, and *W*t is the weight of dried butadiene-isoprene copolymer.

The intrinsic viscosity [η] of the resulting polymers was detected on the basis of the following procedure: The polymer samples were firstly dissolved into the toluene solution at a weight concentration (c) of 1.2 mg∙mL^−1^, and then employing a Ubbelohde viscometer to determine the outflow time of the pure toluene (*t*_0_) and the polymer solution (*t*_n_) at 30 ± 0.2 °C. Ultimately, the intrinsic viscosity of the polymer samples could be obtained using the following equation [35,36]:[η] = 3[(*t*_n_*t*_0_^−1^)^1/3^ − 1]·*c*^−1^(2)

The gel fraction of the resulting polymer was detected by a net hinging method. Approximately 0.25 g (*W*_p_) of sample pieces were packaged into the nickel screen by weight of *W*_0_ and was soaked in 50 mL toluene solution for 48 h at room temperature. The nickel screen was subsequently removed from the toluene solution and dried in vacuum to a constant weight, signed as *W*_1_. The gel fraction was calculated in terms of the following equation:Gel% = (*W*_1_ − *W*_0_)∙100%∙*W*_P_^−1^(3)

The solution viscosity measurements were performed by detecting the polymer solution viscosity at different rotation speeds at 25 ± 0.2 °C using a DV-C NDJ series digital viscometer. The polymer samples were dubbed into a 3.3 g·110 mL^−1^ solution with cyclohexane as the reagent.

The weight-average molecular weight (*M*_w_), number-average molecular weight (*M*_n_) and molecular weight distribution (*M*_w_/*M*_n_) were measured with gel permeation chromatography (GPC) from Waters Instrument (America) at 30 °C using tetrahydrofuran as an eluent at a flow rate of 1.0 mL∙min^−1^ on the basis of the narrow polystyrene standards as the test references. These measurements were performed on a Waters 1525 binary system (Milford, MA, USA) equipped with a Waters 2414 RI detector using four Styragel columns from 1000 e to 1,000,000 Å.

The fourier transform infrared (FTIR) spectra was recorded on a Nicolet Magna 750 spectrometer (Thermo Nicolet Corporation, Madison, WI, USA) in a range of 4000–500 cm^−1^ by total reflection mode. In terms of the absorption peak area, the contents of 1,4- and 1,2-structures could be easily obtained [37,38,39].

The glass transition temperature (*T*_g_) and crystalline behavior were obtained from the differential scanning calorimetry (DSC) curve that was conducted using a TA DSC-Q20 thermal analyzer (New Castle, DE, USA) under a nitrogen atmosphere.

The ^1^H NMR spectra was recorded on a 500 MHZ Bruker Ultra-Shield^TM^ 500 MHz NMR spectrometer (Bruker Ultra-Shield, Karlsruhe, Germany) at room temperature with deuterated chloroform (CDCl_3_) as the solvent and tetramethylsilane (TMS) as an internal chemical shift reference. The microstructure and sequence distribution of polymers could be calculated from the integral area of the characteristic peaks [40].

## 3. Results and Discussion

### 3.1. Effects of Different Ligands on the Copolymerization Activity

Five different compounds, including 2,6-di-tert-butyl-4-methylphenol (264), 2,4-bis(n-octylthiomethyl)-6-methylphenol (2088), (2-ethylhexyl)phosphoric acid (P_204_) and tributyl phosphate (TBP) as the ligands of pentavalent molybdenum catalysts, and tris(nonylphenyl)phosphite (TNPP) as the ligand of the hexavalent molybdenum catalyst, were employed to investigate the effects of types of ligands on the butadiene-isoprene copolymerization activity. It is found that all the ligands exhibited a good catalytic effect, which indicated that these ligands can solubilize main catalyst Mo, commendably. The effects of the five types of ligands on polymerization activity are shown in Table 1 and the corresponding structural formula is given in Figure 1.

It can be concluded from Table 1 that the five types of ligands can display polymerization activity to the butadiene-isoprene copolymerization. However, the TNPP ligand exhibits superior monomer conversion and intrinsic viscosity compared to other four types of ligands, which indicates that the polymerization activity initiated by the ligand TNPP is much lower than that initiated by other ligands. Employing MoO_2_Cl_2_ as the main catalyst and TNPP as the ligand to initiate the butadiene-isoprene copolymerization can achieve higher monomer conversion (>70%). Accordingly, in our following research, the butadiene-isoprene copolymerization experiments are carried out with MoO_2_Cl_2_, phenol-substituted aluminum alkyl with TNPP as the main catalyst, co-catalyst and ligand respectively, and hydrogenated gasoline as the solvent.

#### Microstructure of the Main Catalyst

The microstructures of the main catalyst were characterized by ^31^P NMR spectra and IR spectra. The chemical shift of P in TNPP occurs at 129 ppm and 0.15 ppm, and after the blending of TNPP with MoO_2_Cl_2_, the area of chemical shift at 129 ppm decreased, and the peak at 0.15 ppm broadens. The reason for the broadening of peak at 0.15 ppm is that there may be competitive reactions in the polymerization system. At the same time, a new characteristic peak appears at the chemical shift of −17 ppm, which is due to the coordination of O atoms on TNPP with metal Mo, thus reducing the density of electron cloud beside P, shifting the displacement to a low field, ultimately producing a novel complex TNPP∙MoO_2_Cl_2_, as indicated in Figure 2a. It can be observed from Figure 2b that in the case of the IR spectra of TNPP, there appeared two remarkable characteristic absorption peaks at 1600 cm^−^^1^ and 1450 cm^−^^1^ attributing to the stretching vibration of the benzene ring skeleton, and there appeared a broad and intense characteristic absorption peak at 747 cm^−^^1^ attributing to the C–H out-of-plane bending vibration of benzene ring. The characteristic absorption peak at 960 cm^−^^1^ was assigned to the P–O–C groups. Nevertheless, in the case of the IR spectra of TNPP∙MoO_2_Cl_2_, the peak at 960 cm^−^^1^ (P–O–C stretching vibration) disappeared compared to pure TNPP, accounting for the reason why the coordination between the lone pair electrons of elemental O in TNPP and the d orbital of elemental Mo in MoO_2_Cl_2_ results in the disappearance of the characteristic peak of P–O–C bonds. Meanwhile, the characteristic absorption peak at 1010 cm^−^^1^ demonstrated a weak deviation, and the peak width increased and the peak intensity decreased. Such results clearly show that the coordination of TNPP and MoO_2_Cl_2_ can produce new complexes instead of the simple addition of TNPP and MoO_2_Cl_2_. The NMR spectra was similar to the IR results, which further confirmed the formation of the new complex MoO_2_Cl_2_∙TNPP.

### 3.2. Effect of Proportion of Catalyst Components on the Butadiene-Isoprene Copolymerization

#### 3.2.1. Effect of TNPP Content

In this experimental section, with TNPP-modified MoO_2_Cl_2_ as the main catalyst, and immobilizing the mole ratios of Al:Mo and Mo:[M], the effects of different mole ratios on the monomer conversion and intrinsic viscosity were investigated, as shown in Figure 3.

In this research, in the case of MoO_2_Cl_2_·TNPP/Al(OPhCH_3_)(i-Bu)_2_ catalytic system, the effects of different P:Mo mole ratios on the monomer conversion and polymer intrinsic viscosity were explored. As demonstrated in Figure 3, for the P:Mo mole ratios in the range of 2 to 6, the monomer conversion rate gradually increased and the intrinsic viscosity gradually decreased with an increase in the P:Mo mole ratio. Moreover, when the P:Mo mole ratio was 6, the monomer conversion reached the highest value, approximately 82%, accounting for the reason why the increase in the P:Mo mole ratio could result in more MoO_2_Cl_2_ compounds dissolved into the hydrogenated gasoline and had more active centers produced, thereby improving the monomer conversion rate. Moreover, the increasing amount of active centers can induce the reduction in the polymer molecular weight and the intrinsic viscosity as well.

#### 3.2.2. Effect of Different Al:Mo Mole Ratios

The polymerization activity of a Zeigler-Natta catalyst depends primarily on the types of main and auxiliary catalysts. However, the relative molecular weight between main catalyst and co-catalyst can also exert a certain effect on the polymerization activity in the reaction system. Therefore, it is indispensable to investigate the effects of different mole ratios of main catalyst and co-catalyst on the monomer conversion rate and intrinsic viscosity.

In this experimental section, with MoO_2_Cl_2_ as the main catalyst and TNPP as the ligand, and immobilizing the mole ratios of P:Mo, Mo:[M] and Bd:Ip, the effects of different Al:Mo mole ratios on the monomer conversion and polymer intrinsic viscosity were investigated, as shown in Figure 4.

It can be seen from Figure 4 that as the Al:Mo mole ratios increase, the monomer conversion rate first significantly increased and then slightly decreases, and goes through a maximum at a Al:Mo mole ratio of 5. This is because the increasing Al:Mo mole ratio is beneficial for the significant increment of active centers, thus resulting in the evident improvement in the monomer conversion. However, when the Al:Mo mole ratio increased to 10, the Mo was over reduced by the Al, ultimately leading the monomer conversion to decrease. The polymer intrinsic viscosity gradually decreases with the increase in the Al:Mo mole ratio and attain a relatively constant at a Al:Mo mole ratio > 15, which is attributed to the fact that superfluous Al compounds can produce damaging effects on the active centers of the catalytic system. Simultaneously, the chain transfer of the Al is also enhanced, thereby reducing the relative molecular weight and intrinsic viscosity of the resulting copolymers. Consequently, in the following experiments, a Al:Mo mole ratio of 5 was employed to explore the effects of other polymerization conditions on the monomer conversion rate and intrinsic viscosity.

#### 3.2.3. Effects of Mo:[M] Mole ratio

In this experimental section, with MoO_2_Cl_2_ as the main catalyst, TNPP as the ligand, and immobilizing the mole ratios of P:Mo, Al:Mo and Bd:Ip, the effects of different Mo:[M] mole ratios on the monomer conversion and polymer intrinsic viscosity were investigated, as shown in Figure 5.

It can be seen from Figure 5 that the Mo:M mole ratios ranging from 3 × 10^−4^ to 8 × 10^−4^, the monomer conversion rate first significantly increased and then slightly increased with an increase in the Mo:M mole ratios. The former is assigned to the reason why the increasing Mo:M mole ratio increased the amount of Mo compounds in unit volume and active centers, further improving the reaction activity, therefore, the monomer conversion rate was also increased accordingly; the latter was because excessive amount of catalysts make the number of active centers maintain a relative balance, thus the monomer conversion rate had a slow growth trend. In order to ensure the monomer conversion higher than 80%, the mole ratio of Mo:M should be 8 × 10^−4^_,_ with an 82% monomer conversion rate. However, it can be also seen from Figure 3 that the intrinsic viscosity gradually decreased with the increase in the Mo:[M] mole ratio mainly due to the fact that the increasing amount of active centers resulted in a reduction of the polymers relative molecular weight.

### 3.3. Effect of Monomer Concentration on the Polymerization Activity and Intrinsic Viscosity

It can be concluded from the research on the effects of catalyst component ratios on the butadiene-isoprene copolymerization that when the mass concentration of [M] was 0.18 g∙mL^−1^, and the mole ratios of TNPP:Mo, Al:Mo and Mo:[M] are 6, 5, and 8 × 10^−4^, a high monomer conversion ratio (>80%) and a low intrinsic viscosity can be synchronously satisfied.

Figure 6 demonstrates the effects of monomer concentration on the monomer conversion and intrinsic viscosity of the resulting polymers. The monomer conversion and intrinsic viscosity first increased and subsequently decreased with the increase in monomer concentration, and reached a maximum at a [M] concentration of 0.16 g∙mL^−1^, which was due to the fact that low monomer concentrations can lessen the coordination opportunity between catalyst and monomers, and some catalysts can easily suffer from inactivation due to failure to react with monomers in a timely manner. As the monomer concentration increased, the reaction probability between catalysts and monomers is remarkably enhanced, which is much more beneficial for butadiene-isoprene copolymerization. Therefore, the monomer conversion was accordingly increased. When the weight concentration of [M] was up to 0.16 g∙mL^−1^, the increasing concentration began to depress the monomer conversion, attributed to too high a monomer concentration resulting in the polymerization temperature difficult to control, and leading active centers that tend towards instability and causing some of active centers to be inactive, thereby producing adverse effects on the polymerization activity and decreasing the intrinsic viscosity of the resulting polymers.

Although the improvement in the monomer concentration was favorable to enhance the monomer conversion and intrinsic viscosity, it also increased the addition of butadiene monomers. In consideration of the butadiene polymerization as an exothermal reaction, the increasing monomer concentration can give rise to an increase the polymerization rate, ultimately resulting in the polymerization temperature being difficult to control, which is disadvantageous for reaction safety.

### 3.4. Effect of Butadiene/Isoprene Mole Ratio on Butadiene-Isoprene Copolymerization

#### 3.4.1. Effect of Butadiene/Isoprene Mole Ratio on the Monomer Conversion and Intrinsic Viscosity

As shown in Figure 7, both the monomer conversion and intrinsic viscosity were firstly reduced and then improved with a decrease in the Bd/Ip mole ratio. This is because the increasing addition of isoprene is of great benefit for the stability of the insertion coordination and decreases the number of active centers for the coordination reaction, thus reducing the monomer conversion and intrinsic viscosity. However, when the mole ratio of Bd:Ip is 0:10, that is, when there only exist isoprene monomers in the polymerization system, homopolymerization active centers are far more numerous than copolymerization active centers. Therefore, in the case of a mole ratio of 0:10, the monomer conversion and intrinsic viscosity obviously increase compared to that at 2:8 of the mole ratio. It can be concluded from Figure 7 that the Bd/Ip mole ratio of 9:1 can achieve the highest conversion.

#### 3.4.2. Effect of Butadiene/Isoprene Mole Ratio on the Polymer Solution Dynamic Viscosity

The dynamic viscosity of polymer solution can indirectly reflect the glue fluidity in the polymerizer and pipeline. Therefore, the variation in the shear viscosity of the polymer solution fabricated at different Bd:Ip mole ratios was investigated, as indicated in Figure 8. Figure 8 demonstrates the relationship between shearing rate (γ) and dynamic viscosity (μ) for the resulting polymers fabricated at different Bd:Ip mole ratios from 0:10 to 9:1. As the Bd:Ip mole ratio decreased, the dynamic viscosity of the resulting polymers first decreased and subsequently increased, the variation tendency of which is perfectly consistent with the intrinsic viscosity data from Figure 8. It can be seen from the Table 2 that the molecular weight of the copolymer firstly decreased and then increased with the increase in Bd:Ip mole ratio, which is consistent with the intrinsic viscosity data. The reason why the viscosity of the copolymer firstly decreased and then increased may be that the change of molecular weight of the copolymer results in the change of the viscosity. Meanwhile, the reason may also be explained as the catalytic system exhibits higher polymerization activity to butadiene, and squints towards initiating the homopolymerization of butadiene monomers or isoprene monomers.

#### 3.4.3. Effect of Butadiene/Isoprene Mole Ratio on the Gel Content

The effect of Bd:Ip mole ratios on the gel content of the resulting copolymers are summarized in Table 2. In addition to *n*_Bd_:*n*_Ip_ = 2:8, the gel content of the resulting copolymers fabricated at other Bd/Ip mole ratios is less than 0.06%, indicating that there substantially does not exist branched and crosslinking structures on the molecular chains. At a Bd:Ip mole ratio of 2:8, the gel content of the resulting copolymers can reach 2.41%, which conjectures that the Mo active centers prefer to initiate the butadiene homopolymerization. However, there are some butadiene monomers in the polymerization system, resulting in the active centers more inclined to react with the butadiene structure unites on the molecular chains, thus inducing more crosslinking structures to produce. In the process of industrial production of butadiene-isoprene copolymers, in order to avoid excessive gel content, the method of introducing more butadiene monomers in the early stage of polymerization should be adopted.

#### 3.4.4. FTIR Spectra of the Copolymers

The IR spectra of the resulting copolymers fabricated at the Bd:Ip mole ratios of 0:10, 2:8 and 9:1 respectively is shown in Figure 9 and structural units attribution of copolymers is given in Table 3. It can be seen that in the case of the polyisoprene fabricated by Ip homopolymerization initiated by the MoO_2_Cl_2_·TNPP/Al(OPhCH_3_)(i-Bu)_2_ catalytic system, characteristic absorption bands at 838 cm^−1^, 1376 cm^−1^, 890 cm^−1^ and 910 cm^−1^ were observed, illustrating that there were cis-1,4, 3,4- and 1,2-structures on the molecular chains of polyisoprene.

When introducing Bd monomers into the polymerization system, various characteristic peaks of polyisoprene begin to weaken with the increase in the Bd/Ip mole ratio. Among them, when the Bd/Ip mole ratio is 2:8, there does not only exists remarkable characteristic absorption peaks of polyisoprene, but also exhibit stronger characteristic absorption peaks at 910 cm^−1^ and 990 cm^−1^ attributing to 1,2-structures compared to polyisoprene. Meanwhile, there also appears an evident characteristic absorption peak at 967 cm^−1^ attributing to the trans-1,4 structures. Such results demonstrate that the butadiene units are present in the resulting polymers fabricated at a Bd:Ip mole ratio of 2:8 in the form of 1,2-structures and trans-1,4 structures, namely, the MoO_2_Cl_2_·TNPP:Al(OPhCH_3_)(i-Bu)_2_ can initiate the butadiene-isoprene copolymerization to obtain the butadiene-isoprene copolymers. When the Bd:Ip mole ratio increased to 9:1, the characteristic absorption peaks of polyisoprene hardly exist. However, the peaks at 910 cm^−1^, 990 cm^−1^ and 967 cm^−1^ became stronger, implying that the number of 1,2- and trans-1,4 structures on the molecular chains of the copolymers increased, and no peak at 738 cm^−1^ was observed, illustrating that there did not appear to be cis-1,4 structures on the molecular chains of the resulting copolymers.

#### 3.4.5. ^1^H NMR Spectra of the Copolymers under Low Monomer Conversion

The microstructures of the butadiene-isoprene copolymers initiated by the MoO_2_Cl_2_·TNPP:Al(OPhCH_3_)(i-Bu)_2_ catalytic system were characterized by 500 MHz high-resolution NMR spectra. Table 4 shows the representative ^1^H NMR peaks and their respective assignment for the butadiene-isoprene copolymers. Figure 10 demonstrates the ^1^H NMR spectra of the copolymers initiated by different Bd:Ip mole ratios, the concrete microstructure data of which are concluded in Table 5. Meanwhile, in the case of the copolymers, the reactivity ratios of the butadiene monomers and isoprene monomers are summarized in Table 6.

Various microstructure contents of the butadiene-isoprene copolymers based on ^1^H NMR are obtained by the following equations [33,34]:
*I*_1,2_ = *A*_5.85~5.60_,*B*_1,2_ = (1/2) × *A*_5.02~4.80_,*B*_1,4_ = (1/4) × (2*A*_5.60~5.20_ − *A*_5.02~4.8_),*I*_1,4_ = *A*_5.20~5.02_ − *A*_5.85~5.60_,*I*_3,4_ = (1/2) × *A*_4.8~4.60_(4)

It can be concluded from Figure 10 and Table 5 that as the Bd:Ip mole ratio increased, the content of the butadiene constitutional units in butadiene-isoprene copolymers gradually increased, and the 1,2- and 1,4-structure content in the butadiene units were also gradually improved. However, the content of the 1,2- and 3,4-structure content in the isoprene structure units gradually decreased. The ^1^H NMR results indicated that the butadiene coordination polymerization is primarily based on the trans- coordination, therefore, the 1,2-structures dominate in the molecular chains of the as-prepared copolymers. The isoprene coordination polymerization is mainly based on 1,2- and 3,4-structures, nevertheless, the increasing addition of isoprene monomers can result in the orientation of the MoO_2_Cl_2_·TNPP:Al(OPhCH_3_)(i-Bu)_2_ catalytic system to 1,2- and 3,4-structures significantly reduced.

On account of the ^1^H NMR results of the resulting polymers fabricated by the MoO_2_Cl_2_·TNPP:Al(OPhCH_3_)(i-Bu)_2_ catalytic system at a low conversion rate (<10%), the reactivity data of the butadiene monomers and isoprene monomers are calculated as indicated in Table 6.

The revised KT approach was employed to draw the η-ζ curve to determine the theoretical equation of the reactivity ratio of the monomers (see Figure 11).

As shown in Figure 11, the datapoints obtained at different Bd:Ip mole ratios followed a saturation curve. The straight line obtained from the η-ζ curve can represent the equation *Y*= 1.72*X* − 0.59. It can be comprehended from the straight line that the slope and intercept are *r*_1_ + *r*_2_∙*a*^−1^ and −*r*_2_∙*a*^−1^, respectively. The reactivity ratios of the butadiene monomer and isoprene monomer are *r*_1_ = 1.13 and *r*_2_ = 0.31 respectively, where *r*_1_ > 1 indicates that the butadiene monomers tend to homo-polymerization, *r*_2_ < 1 indicates that the isoprene monomers tend to copolymerize with the butadiene monomers, and *r*_1_*r*_2_ < 1 conjectures that the reaction is geared to the non-ideal non-constant ratio copolymerization under the catalytic system.

## 4. Conclusions

With a modified-MoO_2_Cl_2_ as the primary catalyst, triethyl aluminum substituted by the m-cresol as the co-catalyst and tris(nonyl phenyl) phosphate (TNPP) as a ligand, butadiene-isoprene copolymers were fabricated by butadiene-isoprene copolymerization. When the copolymerization was performed at 60 °C for 6 h, with the M weight concentration of 0.18 g∙mL^−1^ and the TNPP:Mo, Al:Mo and Mo:[M] mole concentration of 6, 5 and 8 × 10^−4^, the monomer conversion could reach 95% and even higher. When conducting the butadiene-isoprene copolymerization, the monomer conversion and intrinsic viscosity firstly increased and subsequently decreased, and the content of the isoprene structure units in the resulting copolymers was gradually improved, where the butadiene units were primarily based on 1,2-structures. When introducing less content of isoprene monomers, the isoprene unites were primarily based on 1,2- and 3,4-structures, and the orientation of active centers to 1,2- and 3,4-structures weakened with the increase in the addition of isoprene monomers. The polymerization products were mainly the high vinyl butadiene-isoprene copolymers, and the reactivity ratios of the butadiene-isoprene copolymerization initiated by the MoO_2_Cl_2_·TNPP/Al(i-Bu)_2_OPhCH_3_ catalytic system were calculated as follows: *r*_Bd_ = 1.13, *r*_Ip_ = 0.31, *r*_Bd_*r*_Ip_ < 1. This demonstrated that as the butadiene monomers tend to homopolymerize, the isoprene monomers tend to copolymerize with the butadiene monomers, and the copolymerization reaction is attributed to the non-ideal non-constant copolymerization under the catalytic system.

## Figures and Tables

**Figure 1 polymers-11-00527-f001:**
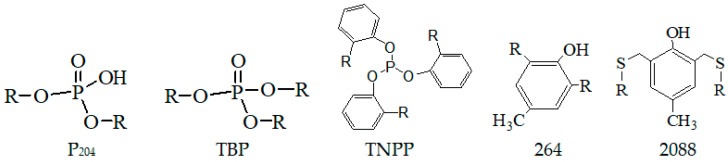
The structural formula of various ligands employed in this research.

**Figure 2 polymers-11-00527-f002:**
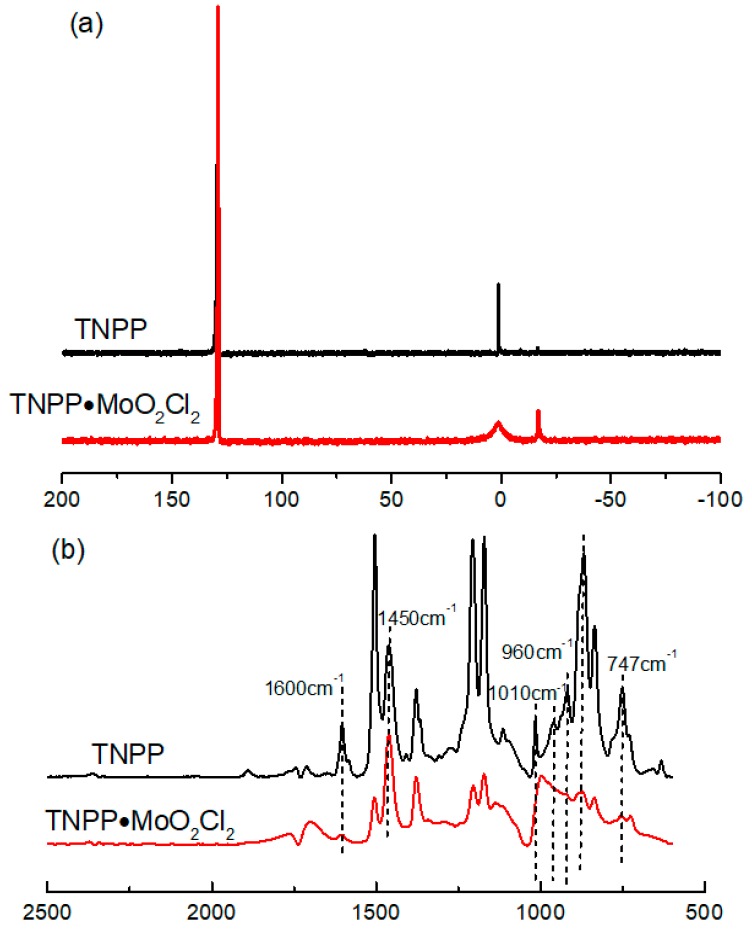
Microstructure characterization of catalysts by ^31^P NMR and FTIR analyses. (**a**) ^31^P NMR spectra of main catalyst. (**b**) IR spectra of main catalyst.

**Figure 3 polymers-11-00527-f003:**
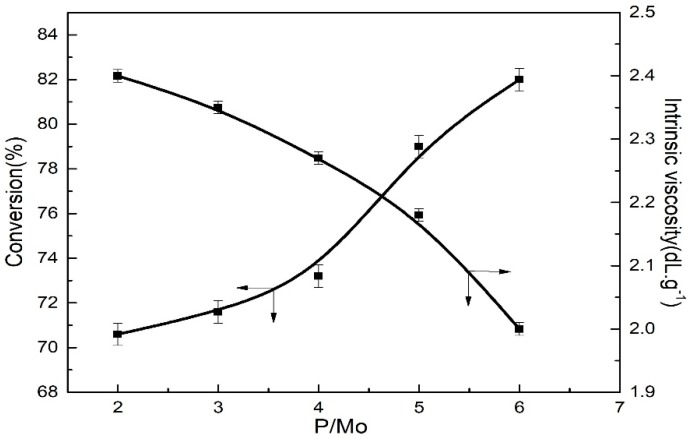
Effect of P:Mo mole ratios from 2 to 6 on the polymerization activity and intrinsic viscosity at a Bd:Ip mole ratio of 9:1. Polymerization conditions: [M] = 0.18 g∙mL^−1^, Mo:[M] = 4 × 10^−4^, Al:Mo = 10, *T* = 60 °C, nBd:nIp = 9:1, *t* = 7 h.

**Figure 4 polymers-11-00527-f004:**
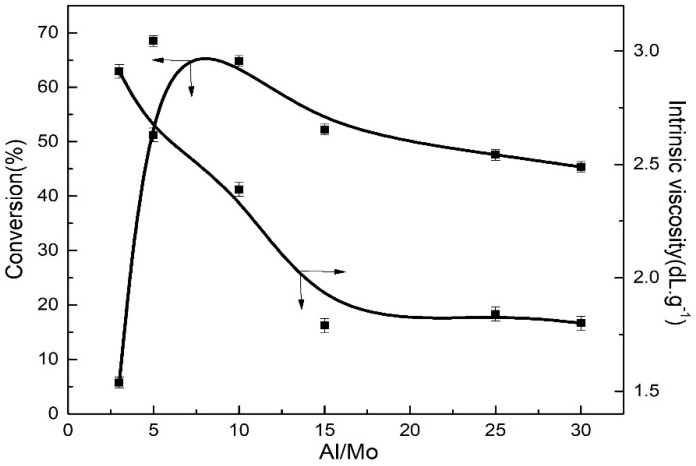
Effect of Al:Mo mole ratio from 3 to 30 on the polymerization conversion and intrinsic viscosity at a Bd:Ip mole ratio of 9:1. Polymerization conditions: [M] = 0.18 g∙mL^−1^, Mo:[M] = 4 × 10^−4^, TNPP:Mo = 6, *T* = 60 °C, nBd:nIp = 9/1, *t* = 7 h.

**Figure 5 polymers-11-00527-f005:**
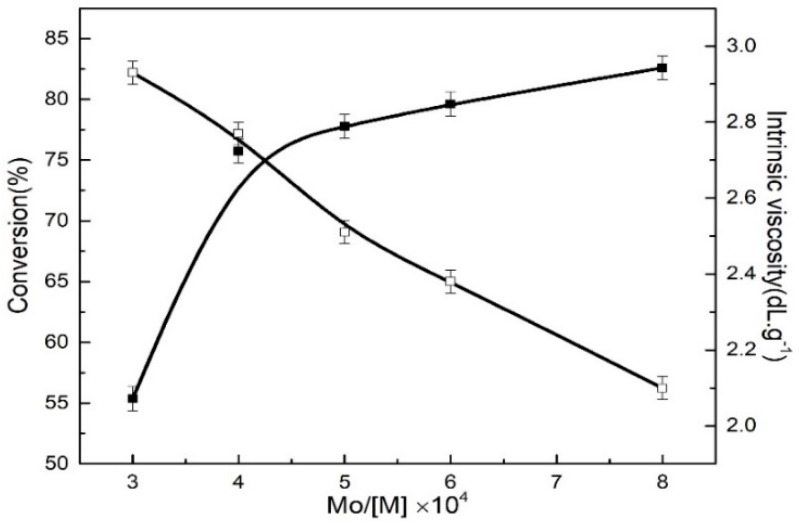
Effect of Mo:M mole ratios from 3 to 8 on the monomer conversion and intrinsic viscosity at a Bd:Ip mole ratio of 9:1. Polymerization conditions: [M] = 0.18 g∙mL^−1^, P:Mo = 6, Al:Mo = 5, nBd:nIp = 9:1, *T* = 60 °C, *t* = 7 h.

**Figure 6 polymers-11-00527-f006:**
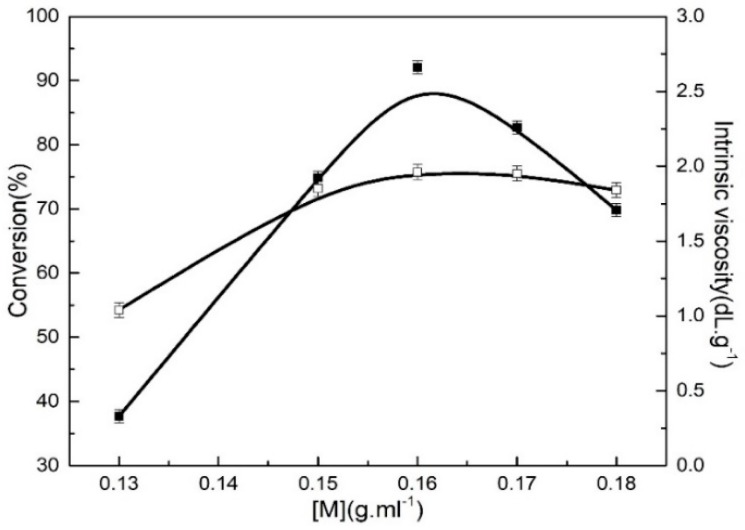
Effect of monomer concentration of 0.13–0.18 g∙mL^−^^1^ on the monomer conversion and intrinsic viscosity at a Bd:Ip mole ratio of 9:1. Polymerization conditions: [M] = 0.18 g∙ml^−1^, TNPP:Mo = 6, Al:Mo = 5, *t* = 7 h, Mo:[M] = 8 × 10^−4^, nBd:nIp = 9:1.

**Figure 7 polymers-11-00527-f007:**
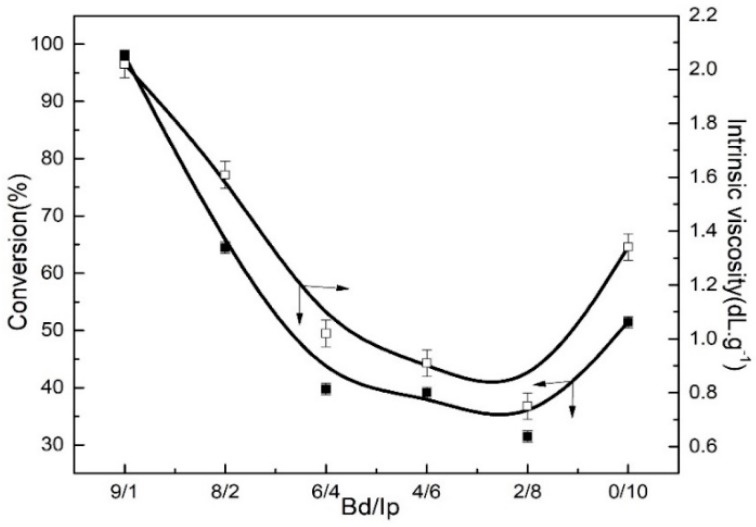
Effect of Bd:Ip mole ratios from 9:1 to 0:10 on the monomer conversion and intrinsic viscosity at a Bd:Ip mole ratio of 9:1. Polymerization conditions: [M] = 0.18 g∙mL^−1^, TNPP:Mo = 6, Al:Mo = 5, *T* = 60 °C, Mo:[M] = 8 × 10^−4^, *t* = 7 h.

**Figure 8 polymers-11-00527-f008:**
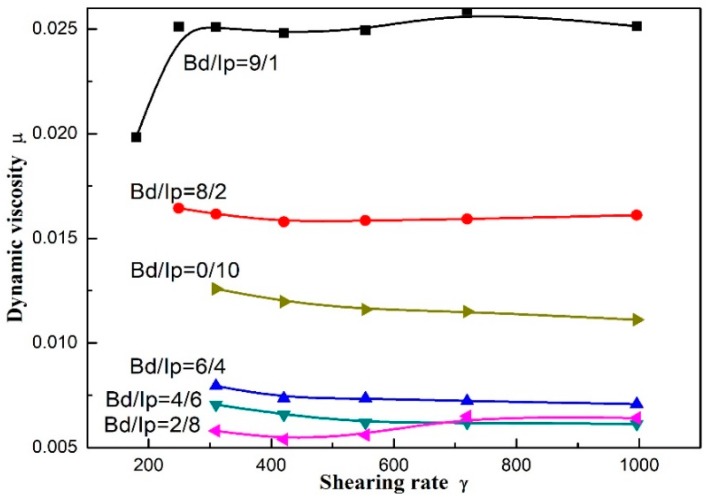
Dynamic viscosity against shearing rate of butadiene-isoprene copolymers synthesized by the coordination polymerization at different Bd:Ip mole ratios from 9:1 to 2:8. Polymerization conditions: TNPP:Mo = 6, Al:[M] = 5, Mo:[M] = 8 × 10^−4^, [M] = 0.18 g∙mL^−1^, *T* = 60 °C, *t* = 7 h.

**Figure 9 polymers-11-00527-f009:**
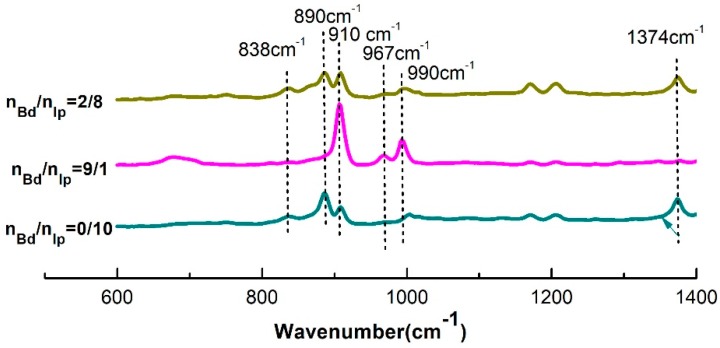
FTIR spectra of the resulting copolymers fabricated by coordination polymerization at the Bd:Ip mole ratios of 2:8, 9:1 and 0:10, respectively. Polymerization conditions: TNPP:Mo = 6, [M] = 0.18 g∙mL^−^^1^, Al:Mo = 5, *T* = 60 °C, Mo:[M] = 8 × 10^−^^4^, *t* = 7 h.

**Figure 10 polymers-11-00527-f010:**
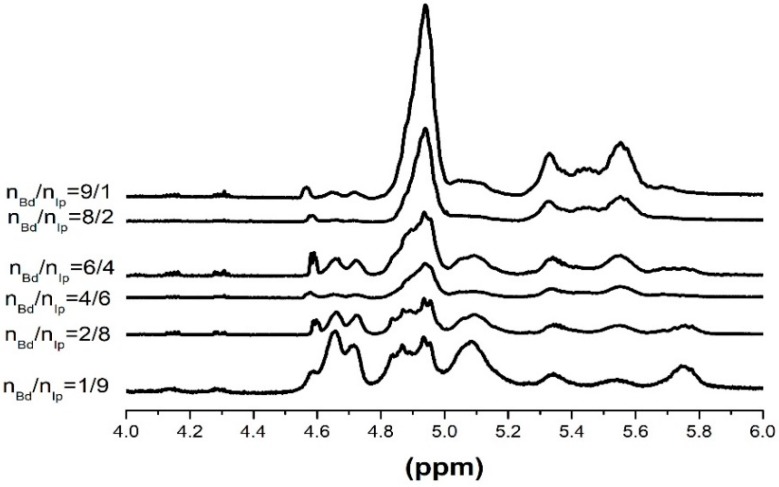
^1^H NMR spectra of the resulting copolymers initiated by different Bd:Ip mole ratios from 9:1 to 1:9. Polymerization conditions: [M] = 0.18 g∙mL^−1^, TNPP:Mo = 6, Al:Mo = 5, *T* = 60 °C, Mo:[M] = 8 × 10^−4^, *t* = 7 h.

**Figure 11 polymers-11-00527-f011:**
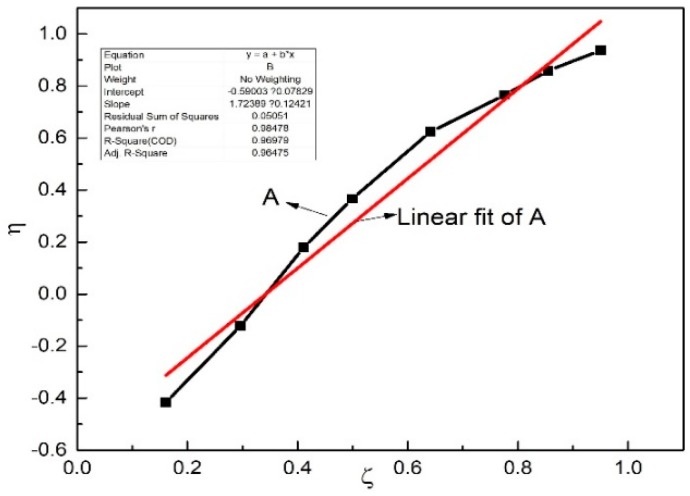
The η-ζ curve of agglomeration rate of the butadiene-isoprene copolymers.

**Table 1 polymers-11-00527-t001:** The effect of different types of ligands on polymerization activity and microstructure.

Different Ligands	Ligands Molecular Formula	Conversion (%)	1,2- %
P_204_	Di(2-ethylhexyl) phosphoric acid (C_16_H_35_O_4_P)	43.2	88.4
TBP	Tributyl phosphate(C_12_H_27_O_4_P)	43.6	85.6
TNPP	Tri (nonylphenyl) phosphite (C_45_H_69_O_3_P)	70.6	84.6
264	2,6-Di-tert-butyl-4-methylphenol (C_15_H_24_O)	47.3	81.6
2088	2,4-Bis(n-octylthiomethyl)-6-methylphenol (C_25_H_44_OS_2_)	25.5	---

Polymerization conditions: Bd/[Ip] = 9/1, ligand/Mo = 2, Mo/[Ip] = 4 × 10^−4^, Al/Mo = 10, *T* = 60 °C, *t* = 7 h.

**Table 2 polymers-11-00527-t002:** Effect of Bd/Ip mole ratios from 9:1 to 0:10 on the gel content of the resulting copolymer.

Bd/Ip	9/1	8/2	6/4	4/6	2/8	0/10
Mn	40.27	42.56	20.84	24.5	22.67	---
PDI	1.90	1.91	1.91	1.90	2.03	---
Gel%	0.06	0	0.22	0.06	2.41	0.06

Polymerization conditions: [M] = 0.18 g∙mL^−1^, TNPP:Mo = 6, Al:Mo = 5, *T* = 60 °C, Mo:[M] = 8 × 10^−4^, *t* = 7 h.

**Table 3 polymers-11-00527-t003:** The wavenumber and corresponding peak assignment of the resulting copolymers obtained from the IR spectra.

Wavenumber/cm^−1^	Assignment
967	Trans-1,4-polybutadiene
910, 990	1,2-polybutadiene
738	Cis-1,4-polybutadiene
838, 1130, 1376	Cis-1,4-polyisoprene
842, 1150, 1385	Trans-1,4-polyisoprene
890	3,4-polyisoprene
910	1,2-polyisoprene

**Table 4 polymers-11-00527-t004:** Representative peaks and corresponding peak assignment of the butadiene-isoprene copolymer.

Number	Data Range (ppm)	Assignment
1	5.85~5.60	Protons of =CH– from non-terminal group in 1,2-isoprene structure
2	5.60~5.20	Protons of –CH=CH– double bonds in 1,4-butadiene structure Protons of –CH= from non-terminal group in 1,2-butadiene structure
3	5.20~5.02	Protons of main chains in 1,4-isoprene structure Protons of CH2= from terminal group in 1,2-isoprene structure
4	5.02~4.80	Protons of CH2= from terminal group in 1,2-butadiene structure
5	4.80~4.60	Protons of CH2= in 3,4-isoprene structure

**Table 5 polymers-11-00527-t005:** Effect of different Bd/Ip mole ratios from 1:9 to 9:1 on the microstructures of the resulting copolymers.

Bd:Ip	*B*_1,2_%	*B*_1,4_%	*I*_1,2_%	*I*_1,4_%	*I*_3,4_%	*B*%	*I*%
1:9	30.82	0	18.90	22.07	28.21	30.82	69.18
2:8	38.18	0	16.99	23.61	21.22	38.18	61.82
3:7	44.37	0.32	16.20	21.48	17.63	44.69	55.31
4:6	52.60	2.11	15.19	16.74	13.36	54.71	45.29
5:5	61.55	4.08	13.21	12.65	8.51	65.63	34.37
6:4	64.39	6.12	11.59	10.55	7.34	70.51	29.49
8:2	74.00	9.69	7.66	5.20	3.48	83.69	16.31
9:1	77.94	11.05	6.69	1.46	2.86	88.99	11.01

**Table 6 polymers-11-00527-t006:** Calculation data for reactivity ratio of Bd and Ip for the resulting copolymers obtained at different Bd:Ip mole ratios from 1:9 to 9:1.

Bd:Ip	*d* _Bd_	*d* _Ip_	*y*	*G*	*F*	*F* ^−1^	G:F	α	η	ζ
1:9	30.823	69.177	0.446	−0.138	0.028	36.091	−4.990	0.527	−0.356	0.050
2:8	38.180	61.820	0.618	−0.155	0.101	9.882	−1.530	0.527	−0.416	0.161
3:7	44.687	55.313	0.808	−0.102	0.227	4.405	−0.449	0.527	−0.240	0.301
4:6	54.712	45.288	1.208	0.115	0.368	2.718	0.312	0.527	0.179	0.411
5:5	65.631	34.369	1.910	0.304	0.524	1.908	0.580	0.527	0.366	0.499
6:4	70.513	29.487	2.391	0.873	0.941	1.063	0.928	0.527	0.624	0.641
7:3	74.87	25.13	2.979	1.716	1.822	0.549	0.942	0.527	0.765	0.776
8:2	83.691	16.309	5.132	3.221	3.118	0.321	1.033	0.527	0.859	0.855
9:1	88.991	11.009	8.083	7.887	10.021	0.100	0.787	0.527	0.937	0.950

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
