# Peer review of "Preparation of Butadiene-Isoprene Copolymer with High Vinyl Contents by Al(OPhCH3)(i-Bu)2/MoO2Cl2∙TNPP"

_polymers, 2019, doi:10.3390/polym11030527_

Round 1

Reviewer 1 Report

The manuscript by Li and co-workers gives insights into the preparation and characterization of copolymers. The work is interesting and the findings are of interest to a broad scientific community including polymer chemists, analytical chemists, and materials scientists. Although this reviewer is supportive of publishing this work in Polymers, the manuscript has several weaknesses. The authors should be able to address the points below during a major revision, after which the manuscript can be further considered for publishing in Polymers.

1. The background provided in the Introduction section is relevant but the discussions are lengthy. Shorten the text to be more concise and sharp.

2. The purity or grade of materials and chemicals should be mentioned under section 2.1.

3. The figure and table cations should be more informative. Currently they are only a few words long and not very descriptive. Add more information so they stand on their own, and facilitate understanding the data presented.

4. Figure 1 and 2 could be combined as a single figure with two panels next to each other.

5. There is no standard deviation / error bars presented for the results in the manuscript. Were the experiments repeated, and were independently prepared polymers used? The authors should prove reproducibility.

6. The authors should briefly mention in the introduction the need for sustainable catalytic systems, and give examples as it is important to stress this trend (ACS Catal., 2018, 8, 7430-7438; J. Catal., 2019, 370, 79-87).

7. Avoid using the ambiguous x/y format for units and apply the x y^-1 instead, which is recommended by the IUPAC.

8. The authors should reconsider the linear fir for the η–ζ curve in Figure 13. Excluding the first point the data seems to follow a saturation curve. More datapoints could be included if available.

9. Potential alternative use of the synthesized copolymers should be mentioned briefly with literature, such as membrane (ACS Nano, 2018, 12, 11471-11480; ACS Appl. Nano Mater., 2018, 1, 6349-6356) and tire-tread (ACS Appl. Nano Mater., 2018, 1, 1003-1008) applications.

10. Some critical assessment should be provided in the manuscript. What are the limitations and drawbacks of the applied methodology? 

11. The reference list is full with errors, typo and missing information. Revise all of them one by one and make sure that the journal’s guideline on formatting is followed. Only first names should be abbreviated, surnames should be spelled out, final page numbers should be added, name of the journals should be abbreviated appropriately.

12. There are several typos and grammatical error throughout the manuscript, serious proofreading by an editing service or native speaker is necessary.

Author Response

1.The background provided in the Introduction section is relevant but the discussions are lengthy. Shorten the text to be more concise and sharp.

RThank you for your comment. In the discussion section of this article, We have deleted some parts of the polymerization conditions, such as reaction time and reaction temperature. This is because the polymerization time and temperature produced little effect on the polymerization activity and intrinsic viscosity of the resulting copolymers in the selected experimental range.

2. The purity or grade of materials and chemicals should be mentioned under section 2.1.

RThank you for your comment. The purity or grade of materials and chemicals has been added in the section 2.1.in Page 4, Line 113-122

3. The figure and table cations should be more informative. Currently they are only a few words long and not very descriptive. Add more information so they stand on their own, and facilitate understanding the data presented.

R: Thank you for your comment. The figure and table captions have been introduced to the manuscript in detail.

4. Figure 1 and 2 could be combined as a single figure with two panels next to each other.

RThank you for your comment. Figure 1 and 2 has been combined and added to the article,as shown in Figure 2. (in Page 7, Line 229-232)

5. There is no standard deviation /error bars presented for the results in the manuscript. Were the experiments repeated, and were independently prepared polymers used? The authors should prove reproducibility.

RThank you for your comment. The error bars has been added in figure 3 (in Page 8, Line 238-239).

6. The authors should briefly mention in the introduction the need for sustainable catalytic systems, and give examples as it is important to stress this trend (ACS Catal., 2018, 8, 7430-7438; J. Catal., 2019, 370, 79-87).  

RThank you for your comment. Organic catalysis has many advantages over traditional transition metal catalysis. Organic catalysis is a more flexible and sustainable catalytic strategy, thus having been an attractive synthesis tool in the catalytic system. The sustainable development of the organic catalysis has been added to the introduction, and the corresponding references have been also added to the revised manuscript (in Page 2, Line 52-55).

18. Zhou, H.; Yang, G.; Zhang, Y.; Xu, Z.; Wu, G. Bio-Inspired Block Copolymer for Mineralized Nanoporous Membrane. ACS Nano, 2018, 12, 11471-11480.

19. Ignacz, G.; Fei, F.; Szekely, G. Ion-Stabilized Membranes for Demanding Environments Fabricated from Polybenzimidazole and its Blends with Polymers of Intrinsic Microporosity. ACS Applied Nano Materials, 2018, 1, 6349-6356.

7. Avoid using the ambiguous x/y format for units and apply the x y^-1 instead, which is recommended by the IUPAC. 

RThank you for your comment. The format of the article has been revised.

8. The authors should reconsider the linear fir for the η–ζ curve in Figure 13. Excluding the first point the data seems to follow a saturation curve. More datapoints could be included if available.

RThank you for your comment. Three datapoints in the η-ζ curve for the butadiene-isoprene copolymers obtained at the Bd:Ip mole ratios of 3:7, 5:5 and 7:3 respectively, and the first datapoint for the butadiene-isoprene copolymers obtained at the Bd:Ip molar ratios of 1:9 was deleted in Figure 13. We redrew the η-ζ curve of agglomeration rate and observed that the datapoints follow a saturation curve. Moreover, we recalculated the reactivity ratios of butadiene and isoprene monomer. The reactivity ratios of butadiene monomer and isoprene monomer were 1.13 and 0.31, respectively (in page 18 Line 459-461).

9. Potential alternative use of the synthesized copolymers should be mentioned briefly with literature, such as membrane (ACS Nano, 2018, 12, 11471-11480; ACS Appl. Nano Mater., 2018, 1, 6349-6356) and tire-tread (ACS Appl. Nano Mater., 2018, 1, 1003-1008) applications.

RThank you for your comment. Copolymer composites are considered as a perspective functional material in various application fields. Selfassembled block copolymer (BCP) stands out as a platform for developing the next-generation high-performance membranes. The application of copolymers has been added to the introduction of the article, and the corresponding references (ACS Nano, 2018, 12, 11471-11480; ACS Appl. Nano Mater., 2018, 1, 6349-6356; ACS Appl. Nano Mater., 2018, 1, 1003-1008) have been also introduced in the revised manuscript (in Page 3, Line 52-55).

10. Some critical assessment should be provided in the manuscript. What are the limitations and drawbacks of the applied methodology? 

RThank you for your comment. In the manuscript, the solution polymerization method was employed during the experiment. Although the operation procedure of the solution polymerization was complex, this method made the viscosity of the polymerization system low, reduced the gel effect, avoided partial overheating, and lowered the risk coefficient of the reaction (in page 3, Line 101-103).

11. The reference list is full with errors, typo and missing information. Revise all of them one by one and make sure that the journal’s guideline on formatting is followed. Only first names should be abbreviated, surnames should be spelled out, final page numbers should be added, name of the journals should be abbreviated appropriately.

RThank you for your suggestion. All the references in this manuscript have been re-checked and the. errors, typo and missing information have been revised.

12. There are several typos and grammatical error throughout the manuscript, serious proofreading by an editing service or native speaker is necessary.  

RThank you for your suggestion. Several typos and grammatical error have been detailedly corrected in this revised manuscript.

Reviewer 2 Report

Polymers 456757

Title: “Preparation of Butadiene-Isoprene Copolymer with high vinyl contents by Al(OPhCH3)(i-Bu)2/MoO2Cl2·TNPP”

In this paper, the authors show a study of the butadiene-isopropene polymerization, with a Mo compound as catalytic-system. The MoO2Cl2 is the primary catalyst, triethyl aluminum substituted by m-cresol and tris(nonyl phenyl) phosphate as ligand. The conversion is 90%. The ratios butadiene/isopropene are 1.09 and 1.27, respectively.

In my opinion even though correctly performed. The chemistry of this paper is sufficiently interesting, but is necessary one major revision of all manuscript.

Comments:

1)      In the 31P NMR spectrum, the peak af TNPP·MoO2Cl2 appear at same position that TNPP (Fig. 1).

The authors comment the formation of novel complex, observed in 31P NMR spectrum.

The authors would have to complete this study and change Fig. 1.

2)      Comment with major detail the IR spectrum of the TNPP ligand and TNPP·MoO2Cl2 complex. Which are the bands but significant and compare the position and intensity of the bands.

3)      Line 363-365

Revise and re-write the paragraph: “The reason may be····isoprene monomers”

Other comments:

1)      Line 55

Delete: central

2)      Line 57

Delete: reaction

3)      Line 74

CHANGE      [4]and             FOR    [4] and

4)      Line 77

Delete: polymerization

5)      Line 79-80

CHANGE      He group         FOR    He et al.

Add references

6)      Line 84

CHANGE      chains[25]       FOR    chains [25]

CHANGE      Nath[26]         FOR    Nath [26]

Revise all manuscript

7)      Line 85, 86

CHANGE      PH3P               FOR    P(Ph3)3

Revise all manuscript

8)      Line 91

CHANGE      structures[27-30]        FOR    structures [27-30]

9)      Line 95

CHANGE      3,4- stereoselective     FOR    3,4-stereoselective

10)  Line 104

CHANGE      4 A                  FOR    4 Å

11)  Line 141

CHANGE      equation[31, 32]         FOR    equation [31, 32]

12)  Line 161

CHANGE      obtained[33, 35]         FOR    obtained [33, 35]

13)  Line 168, 169

CHANGE      peaks[36,37]               FOR    peaks [36,37]

14)  Line 188

CHANGE      MoC2Cl2                    FOR    MoO2Cl2

15)  Line 189

CHANGE      butadiene                    FOR    butadiene

16)  Line 221

CHANGE      2-6                              FOR    2:6

17)  Line 281

CHANGE      concerntration            FOR    concentration

18)  Line 388

CHANGE      9:1 respectively          FOR    9:1, respectively

19)  Line 391

CHANGE      34400px-1                    FOR    1376 cm-1

20)  Line 396

CHANGE      22750px-1 and 24750px-1 FOR    910 cm-1 and 990 cm-1

21)  Line 441

CHANGE      1H NMR                    FOR    1H NMR

22)  References

[6] [10] [12] [23]

Change           Polymer                      FOR    Polymers

Revise the presentation of the references:

[1] [16] [22] [25] [26] [28] [29] [37]

Author Response

1)      In the 31P NMR spectrum, the peak af TNPP·MoO2Cl2 appear at same position that TNPP (Fig. 1).

The authors comment the formation of novel complex, observed in 31P NMR spectrum.

The authors would have to complete this study and change Fig. 1.

RThank you for your suggestion. We have re-performed the measurement of 31P and the figure has been modified in the revised manuscript. The microstructures of the main catalyst were characterized by 31P NMR spectra and IR spectra. The chemical shift of P in TNPP occurs at 129 ppm and 0.15ppm, and after the blending of TNPP with MoO2Cl2, the area of chemical shift at 129 ppm decreases, and the peak at 0.15 ppm broadens. The reason for the broadening of the peak at 0.15 ppm is that there may exist various competitive reactions in the polymerization system. At the same time, a new characteristic peak appears at the chemical shift of -17 ppm, which is due to the coordination of P atoms in TNPP with the metal Mo, thus reducing the density of electron cloud beside P, and shifting the displacement to low field, further producing a novel complex TNPP• MoO2Cl2, as indicated in Figure 2.(in page 7 Line 209-216).

2) Comment with major detail the IR spectrum of the TNPP ligand and TNPP·MoO2Cl2 complex. Which are the bands but significant and compare the position and intensity of the bands.

R: Thank you for your comment. We have re-made a detailed analysis on the IR spectra. The corresponding explaination is indicated as follows:

    It can be observed from Figure 2(b) that in the case of the IR spectra of TNPP, there appear two remarkable characteristic absorption peaks at 1600 cm-1 and 1450 cm-1 attributing to the stretching vibration of the benzene ring skeleton, and there appears a broad and intense characteristic absorption peak at 747 cm-1 attributing to the C-H out-of-plane bending vibration of benzene ring. The characteristic absorption peak at 960 cm-1 is assigned to the P-O-C groups. Nevertheless, in the case of the IR spectra of TNPP•MoO2Cl2, the peak at 960 cm-1 (P-O-C stretching vibration) disappears compared to pure TNPP, accounting for the reason why the coordination between the lone pair electrons of O element in TNPP and the d orbital of Mo element in MoO2Cl2 results in the disappearance of the characteristic peak of P-O-C bonds. Meanwhile, the characteristic absorption peak at 1010 cm-1 demonstrates a weak deviation, and the peak width increases and the peak intensity decreases. Such results certifies that the coordination of TNPP and MoO2Cl2 can produce new complexes instead of the simple addition of TNPP and MoO2Cl2. The NMR spectra is similar to the IR results, which further confirmed the formation of the new complex MoO2Cl2•TNPP. (in page 7, Line 216-229).

3)      Line 363-365

Revise and re-write the paragraph: “The reason may be····isoprene monomers”

RThank you for your comment. The paragraph has been re-written as follows.

It can be seen from the table 2 that the molecular weight of the copolymer firstly decreases and then increases with the increase in Bd/Ip molar ratios, which is consistent with the intrinsic viscosity data. The reason why the viscosity of the copolymer first decreases and then increases may be that the change of molecular weight of the copolymer results in the change of the viscosity of. Meanwhilethe reason may be also explained as that the catalytic system exhibits higher polymerization activity to butadiene, and squints towards initiating the homopolymerization of butadiene monomers or isoprene monomers (in Page 13, Line 364-370).

Other comments:

1)      Line 55

Delete: central

2)      Line 57

Delete: reaction

3)      Line 74

CHANGE      [4]and             FOR    [4] and

4)      Line 77

Delete: polymerization

5)      Line 79-80

CHANGE      He group         FOR    He et al.

Add references

6)      Line 84

CHANGE      chains[25]       FOR    chains [25]

CHANGE      Nath[26]         FOR    Nath [26]

Revise all manuscript

7)      Line 85, 86

CHANGE      PH3P               FOR    P(Ph3)3

Revise all manuscript

8)      Line 91

CHANGE      structures[27-30]        FOR    structures [27-30]

9)      Line 95

CHANGE      3,4- stereoselective     FOR    3,4-stereoselective

10)  Line 104

CHANGE      4 A                  FOR    4 Å

11)  Line 141

CHANGE      equation[31, 32]         FOR    equation [31, 32]

12)  Line 161

CHANGE      obtained[33, 35]         FOR    obtained [33, 35]

13)  Line 168, 169

CHANGE      peaks[36,37]               FOR    peaks [36,37]

14)  Line 188

CHANGE      MoC2Cl2                    FOR    MoO2Cl2

15)  Line 189

CHANGE      butadiene                    FOR    butadiene

16)  Line 221

CHANGE      2-6                              FOR    2:6

17)  Line 281

CHANGE      concerntration            FOR    concentration

18)  Line 388

CHANGE      9:1 respectively          FOR    9:1, respectively

19)  Line 391

CHANGE      34400px-1                    FOR    1376 cm-1

20)  Line 396

CHANGE      22750px-1 and 24750px-1 FOR    910 cm-1 and 990 cm-1

21)  Line 441

CHANGE      1H NMR                    FOR    1H NMR

22)  References

[6] [10] [12] [23]

Change           Polymer                      FOR    Polymers

R: Thank you for your comment. We apologize for our carelessness. We have revised these mistakes following your comment. Further, we rechecked all throughout our paper and revised the spelling mistakes.

Revise the presentation of the references:

[1] [16] [22] [25] [26] [28] [29] [37]

RThank you for your suggestion. We further rechecked all the references and revised the errors.

Reviewer 3 Report

In this contribution, the authors describe the copolymerization of butadiene and isoprene promoted by a MoO2Cl2/Al(OPhCH3)Et2/ligand catalytic system. Monomers conversion was dependent on the chosen ligand (tri(nonylphenyl)phosphite-TNPP-was found to be the best) and on several parameters like Al/Mo ratio and catalytic loading. Copolymers viscosity was determined and IR and NMR characterizations were also provided.

Even if the novelty is minor and the results are foreseeable, this referee values the methodical work done and thus recommends publication on Polymers albeit only after resolution of the following points:

-Viscosity gives no information on molecular weight distributions, which are very informative in homo- and copolymerization. For this reason, samples should be characterized also through Gel Permeation Chromatography.

-Reactivity ratios of the monomers were determined and it is clear that we are not dealing with a random copolymerization. So the copolymers should not be defined as poly(butadiene-co-isoprene).

Minor issues:

-Line 177, the adjective excellent should be replaced.

-It would be more advisable to provide a figure for the ligands formula, instead of inserting them in Table 1.

-Many language errors can be found (for example: line 29, have arouse (have aroused); line 55, diolephin (diolefin); line 158, spectra was (spectra were); Table 4, prontons (protons) and many others. A careful English proofreading is recommended.

-Line 116 and 117, cocatalyst and catalyst formula should be corrected.

-In all references, authors surnames should be in full (See, for example, ref. 25).

Author Response

-Viscosity gives no information on molecular weight distributions, which are very informative in homo- and copolymerization. For this reason, samples should be characterized also through Gel Permeation Chromatography.

RThank you for your suggestion. The butadiene-isoprene copolymers were characterized by the gel permeation chromatography (GPC). The result and explaination were demonstrated as follows:

    It can be seen from the table 2 that the molecular weight of the copolymer firstly decreases and then increases with the increase of Bd/Ip, which is consistent with the intrinsic viscosity data. The reason why the viscosity of the copolymer firstly decreases and then increases may be that the change of molecular weight of the copolymer results in the change of the viscosity. Meanwhilethe reason may be also explained as that the catalytic system exhibits higher polymerization activity to butadiene, and squints towards initiating the homopolymerization of butadiene monomers or isoprene monomers. (in Page 13, Line 364-370).

-Reactivity ratios of the monomers were determined and it is clear that we are not dealing with a random copolymerization. So the copolymers should not be defined as poly(butadiene-co-isoprene). 

RThank you for your suggestion. The names of all copolymers in this paper have been corrected.

Minor issues:

-Line 177, the adjective excellent should be replaced.

RThank you for your suggestion. We have used the adjective “good” instead of the adjective “excellent” in Line 187.

-It would be more advisable to provide a figure for the ligands formula, instead of inserting them in Table 1.   

RThank you for your suggestion. We have provide the ligands formula in Figure 1. (in Page 6, Line 193-195)

Figure 1. The Structural formula of Ligands

-Many language errors can be found (for example: line 29, have arouse (have aroused); line 55, diolephin (diolefin); line 158, spectra was (spectra were); Table 4, prontons (protons) and many others. A careful English proofreading is recommended.  

RThank you for your suggestion. We have revised these mistakes following your comment. Further, we rechecked all throughout our paper and revised the spelling mistakes.

-Line 116 and 117, cocatalyst and catalyst formula should be corrected. 

RThank you for your suggestion. We have corrected the above mistakes in my article. The m-cresol-substituted alkyl aluminum was prepared according to the following procedure: a certain volume of AlEt3 is introduced into the reactor with an electromagnetic agitator under the protection of nitrogen atmosphere, and then adding the m-cresol into the triethyl aluminum at a molar ratio of AlEt3/HOPhCH3 of 1:1 to prepare Al(OPhCH3)Et2 co-catalysts.( in Page 4, Line 122-125)

-In all references, authors surnames should be in full (See, for example, ref. 25).

RThank you for your suggestion. We have corrected authors surnames in all references following your comment.

 -Reactivity ratios of the monomers were determined and it is clear that we are not dealing with a random copolymerization. So the copolymers should not be defined as poly(butadiene-co-isoprene).  

RThank you for your suggestion. We have used butadiene-isoprene copolymers instead of poly(butadiene-co-isoprene) in the revisedmanuscript.

Round 2

Reviewer 1 Report

The authors have done a thorough revision but there are some minor editing issues to be correct prior to publication:

Correct 'yiPreparation' -> 'Preparation' in the title.

Correct 'as a perspective functional material' -> 'as perspective functional materials' in line 53.

Correct 'to the asymmetric' to 'to asymmetric' in line 80.

Figure 1 needs to have the chemical structures drawn following the same editing style.

Correct the subscripts in line 227.

There are 2 journal names in Reference 4; correct as necessary (delete extra journal name).

Two names have errors in Reference 25; correct as necessary (Barabas, J.; Vass, E.;).

Remove city names from the journals in References 20-22.

Author Response

We sincerely appreciate your carefully reading, very helpful comments and suggestions. We have discussed the comments carefully and revised the manuscript accordingly. Detailed corrections are listed below point by point. All contents revised and new additions in this updated manuscript have been highlighted in red font for your identification easily.

Correct 'yiPreparation' -> 'Preparation' in the title.

R:Thank you for your comment. I have revised the title (Page 1, Line 1).

Correct 'as a perspective functional material' -> 'as perspective functional materials' in line 53.

R:Thank you for your comment. I have revised it in line 53.

Correct 'to the asymmetric' to 'to asymmetric' in line 80.

R:Thank you for your comment. I have revised it in line 79.

Figure 1 needs to have the chemical structures drawn following the same editing style.

R:Thank you for your comment. I have redrawn the chemical structures following the same editing style (Page 6, Line195-197).

Correct the subscripts in line 227.

R:Thank you for your comment. I have revised the subscripts in line 227.

There are 2 journal names in Reference 4; correct as necessary (delete extra journal name).

R:Thank you for your comment. I have revised the wrong journal name in Reference 4.

Two names have errors in Reference 25; correct as necessary (Barabas, J.; Vass, E.;).

R:Thank you for your comment. I have revised the inaccurate author names in Reference 25.

Remove city names from the journals in References 20-22.

R:Thank you for your comment. I have removed the city names from the journals in References 20-22.

Reviewer 2 Report

In my opinion, the paper can be accepted in present form

Author Response

We sincerely appreciate your carefully reading, very helpful comments and suggestions. Meanwhile, we are very grateful for your recongnition and support of our revised manuscript